# Pediatric Spinal Cord Injury: A Review

**DOI:** 10.3390/children10091456

**Published:** 2023-08-26

**Authors:** Natalia S. C. Cunha, Anahita Malvea, Sarah Sadat, George M. Ibrahim, Michael G. Fehlings

**Affiliations:** 1Hospital da Criança de Brasília, Brasília 70684-831, Brazil; nataliaspinola@gmail.com; 2Division of Neurosurgery, Krembil Neuroscience Centre, University Health Network, Toronto, ON M5T 2S8, Canada; anahita.malvea@mail.utoronto.ca; 3Institute of Medical Science, University of Toronto, Toronto, ON M5S 1A1, Canada; sarah.sadat@mail.utoronto.ca; 4Division of Neurosurgery, The Hospital for Sick Children, Toronto, ON M5G 1E8, Canada; george.ibrahim@sickkids.ca; 5Division of Neurosurgery, Department of Surgery, University of Toronto, Toronto, ON M5S 1A1, Canada

**Keywords:** spinal cord injury, spinal cord dysfunction, traumatic spinal cord injury, non-traumatic spinal cord injury, rehabilitation, neuroregeneration

## Abstract

A spinal cord injury (SCI) can be a devastating condition in children, with profound implications for their overall health and quality of life. In this review, we aim to provide a concise overview of the key aspects associated with SCIs in the pediatric population. Firstly, we discuss the etiology and epidemiology of SCIs in children, highlighting the diverse range of causes. We explore the unique anatomical and physiological characteristics of the developing spinal cord that contribute to the specific challenges faced by pediatric patients. Next, we delve into the clinical presentation and diagnostic methods, emphasizing the importance of prompt and accurate diagnosis to facilitate appropriate interventions. Furthermore, we approach the multidisciplinary management of pediatric SCIs, encompassing acute medical care, surgical interventions, and ongoing supportive therapies. Finally, we explore emerging research as well as innovative therapies in the field, and we emphasize the need for continued advancements in understanding and treating SCIs in children to improve their functional independence and overall quality of life.

## 1. Introduction

Spinal cord injuries (SCIs) in the pediatric population present a unique and complex set of challenges for clinicians, researchers, and families. Every year, a considerable number of children sustain SCIs due to various causes, including trauma, congenital malformations, and acquired diseases [1]. These injuries have profound implications on a child’s overall health, functional abilities, and quality of life. Thus, understanding the complexity of pediatric SCIs and developing effective interventions are of utmost importance [2,3,4].

A pediatric SCI differs from its adult counterpart in several key aspects. Anatomically, the developing pediatric spinal cord exhibits distinct structural characteristics with different response mechanisms to injury [1,5]. The pediatric spinal cord is more pliable and elastic, leading to different injury patterns and the potential for recovery compared to adults. Moreover, children’s growing bodies require unique considerations when planning treatment strategies to ensure optimal functional outcomes while minimizing long-term complications.

Over the years, substantial progress has been made in advancing our knowledge and understanding of pediatric SCI. Medical advancements, improved diagnostic techniques, and multidisciplinary care approaches have contributed to enhanced outcomes and quality of life for children with SCIs [6]. However, numerous challenges and unanswered questions still exist, necessitating further research and innovation.

This manuscript aims to comprehensively explore the current state of knowledge surrounding pediatric SCIs, highlighting recent advancements, unresolved challenges, and promising interventions. By critically examining the existing literature, we aim to provide a comprehensive overview of the unique aspects of pediatric SCIs and their clinical manifestations, underlying mechanisms, and potential therapeutic strategies.

This manuscript delves into various aspects of pediatric SCIs, including epidemiology, etiology, classification, and prognostic factors. We explore the acute management of spinal cord injuries in children, encompassing pre-hospital care, emergency stabilization, and surgical interventions. Additionally, we review the comprehensive rehabilitation approaches, including physical therapy and occupational therapy. 

## 2. Methods

This study employed a combined scoping and narrative review methodology. Our approach included a comprehensive electronic database search across PubMed, PubMed Central (PMC), Scopus, Google Scholar, and the Cochrane Library. To ensure a thorough exploration of the topic, we employed a variety of search terms, namely “Children OR Pediatric AND Spinal Cord Injury”, “Pediatric AND Spinal Cord Injury AND Surgery OR clinical treatment”, “Children AND Spinal Cord Injury AND early surgery”, “Children OR Pediatric AND Spinal Cord Injury AND ICU management OR complications management”, “Children OR Pediatric AND rehabilitation”, and “Spinal cord injury AND new trials OR future treatment”. Subsequently, we retrieved the identified articles and applied a set of rigorous inclusion criteria.

The inclusion criteria encompassed studies that were published in the English language, spanned various primary study designs and trials, and covered the spectrum of the literature from the databases’ inception to the review’s cutoff date (15 February 2023). Our assessment extended to studies examining any form of pediatric spinal cord injury, irrespective of its origin (e.g., traumatic, non-traumatic, and congenital). The scope of eligible studies encompassed diverse outcomes pertaining to pediatric spinal cord injuries, including but not limited to clinical presentations, treatment modalities, strategies for rehabilitation, long-term repercussions, quality-of-life assessments, and complications arising from the condition. 

## 3. Pediatric Spine Anatomy 

The anatomy of the pediatric spine is distinct from adults due to ongoing skeletal maturation and central nervous system (CNS) development. There are numerous differences that must be noted between the adult and pediatric spinal cord, as they have important implications for the management of pediatric SCI. Briefly, this section of the review goes over the key radiographic features that differentiate the pediatric spine from that of adults, as well as the impact of skeletal maturation and CNS development on the spinal cord. 

### 3.1. Contrast to Adults 

The emergency radiologic evaluation of pediatric spine cases can be taxing due to the wide range of normal anatomic variants and injuries that are unique in the pediatric population. Thus, it is important to understand the key differences that occur during normal embryonic development to avoid misinterpretations, as well as improving accuracy in image interpretation. To begin, the pediatric spine is made of 33 vertebrae, including 7 cervical, 12 thoracic, 5 lumbar, 5 sacral, and 4 coccygeal vertebrae. This contrasts with the adult spine, where the number is reduced to 26 vertebrae, as some vertebrae fuse during normal growth and development [1,7]. The anatomy of the spinal cord starts to resemble the adult spine by the age of 8–10 years. Importantly, in children, the center of rotation (COR) is shifted as compared to adults. Around the age of 8–10 years old, the COR of the cervical spine is located at the C5–6 level; however, prior to this age, the COR is located at C2–3. This shift is partially responsible for the increase in upper cervical spine injuries in children compared to adults [1]. 

Another key difference is that children have larger volumes of total and spinal cerebrospinal fluid (CSF), with 50% in children versus only 33% in adults [8]. In addition, spinal ligaments are less densely packed together; thus, there is increased spine flexibility [9,10]. The pediatric spine is considered hypermobile compared to that of adults due to many differences, including the shallow and angled facet joints, increased ligamentous laxity, and underdeveloped spinous processes [11,12]. Furthermore, younger children may also have incomplete ossification of the odontoid process extending from the C2 level, and along with a relatively large head and weaker neck muscles, this increases the risk of spinal instability compared to adults [13,14]. 

### 3.2. Skeletal Maturation

Skeletal maturation can significantly influence the anatomy of the pediatric spine and, thus, can influence the degree of the SCI depending on the level of maturation that has occurred. For instance, in children whose cervical spine has not yet undergone ossification, SCIs are more likely to occur at lower levels than in adults. The formation of the axial skeleton begins during early embryonic development. Around 5-to-6 weeks of gestation, the vertebral bodies of the spinal cord undergo chondrification; however, growth, remodeling, and ossification persist for decades after birth before the complete development of the adult skeleton [15].

Many studies have investigated the timing and pattern of skeletal maturation in the pediatric spine. Ossification centers first appear in the developing spine for the neural arches of the cervical and upper thoracic vertebrae. This process begins during the embryonic period and continues throughout fetal development. Using magnetic resonance imaging (MRI) to assess vertebral ossification in children, it is generally concluded that ossification begins in fetuses at around 10–11 weeks [16]. By the 10th week of gestation, ossification centers for vertebral centra are present in the lower seven thoracic and first lumbar vertebrae. By the end of the 11th week, ossification centers for vertebral centra are present in the lower four cervical, all thoracic, all lumbar, and four sacral vertebrae. The pattern of ossification for neural arches proceeds in a craniocaudal direction, while in vertebral centra, it progresses from the lower thoracic vertebrae in both directions. Between the ages of 3 and 6 years, the neural processes fuse with the centrum, and during puberty, five secondary ossification centers are formed at the top of the spinous processes on both surfaces of the vertebral body that are responsible for the superior–inferior growth of the vertebrae. This process of ossification continues to occur until adulthood and ends around the age of 18–25 [1]. 

### 3.3. CNS Development 

The development of the spinal cord in children is ongoing and affects the anatomy of the pediatric spine. In children, the spinal cord is more vulnerable to injury due to its greater blood supply, which can lead to more extensive secondary injury following an SCI. Additionally, the immature CNS of children may not be able to respond as effectively to injury as in adults, possibly resulting in worse functional outcomes [17]. 

The development of the CNS of the pediatric spine involves multiple complex stages, including neural tube formation, neurogenesis, axon guidance, and synaptogenesis, as well as myelination. The first stage of CNS development in the pediatric spine is the formation of the neural tube, which occurs during the first four weeks of gestation. The neural tube gives rise to the brain and spinal cord and is formed by the folding of the neural plate. The neural tube is closed in a rostral-to-caudal direction, and defects in this process can result in neural tube defects such as spina bifida, where the spinal column is not completely closed [4]. Once the neural tube is formed, neurogenesis begins. This process involves the proliferation and differentiation of neural stem cells into neurons and glial cells. In the spinal cord, neurogenesis occurs primarily in the ventricular zone, which is located near the central canal. The timing and extent of neurogenesis are tightly regulated and play a critical role in the formation of the proper number and types of neurons, as well as the glia, in the developing spinal cord [18]. 

The next stage involves the formation of synapses and axon guidance. As neurons begin to differentiate, they extend axons to their targets and form synapses. Axon guidance and synaptogenesis are highly regulated processes that involve the interaction between growth cones on the tips of axons and guidance cues in the environment. These extracellular guidance cues include secreted factors, such as netrins and semaphorins, as well as cell adhesion molecules (CAMs), such as cadherins and ephrins [19]. In the developing spinal cord, axon guidance and synaptogenesis are critical for the formation of functional circuits that allow for sensory and motor function. Lastly, the final stage of CNS development in the pediatric spine is myelination, which begins in the third trimester of gestation and continues into early adulthood. Myelination involves the formation of myelin sheaths around axons by oligodendrocytes. This process is critical for proper nerve conduction and allows for the efficient transmission of signals in the spinal cord [8]. 

## 4. Pediatric SCI Epidemiology

This section focuses on the epidemiology of SCI in pediatric populations, including a review of its incidence, etiology, and pathophysiology.

### 4.1. Incidence

Depending on the population studied and the definition of SCI used, the incidence of pediatric SCI will vary. According to the National Spinal Cord Injury Statistical Center, spinal cord injury in children is relatively rare and represents less than 4% of the overall SCI incidence annually [20]. In the United States, in particular, the annual incidence of SCI is approximately 54 cases per million, with a higher incidence in males than in females [20]. The incidence of SCI in pediatric populations has also drastically changed over time. In adolescents, the incidence of SCI dramatically decreased from 13 per million people to 8 per million from 1997 to 2012.

With age, the incidence increases rapidly, where injuries at ages 17–23 represent more than 30% of injuries, and 16–30 represents 53% of injuries [21]. Importantly, the rate of recovery after injury is significantly faster in the pediatric population compared to adults. Depending on age, the level of the SCI also varies, with preteen groups prevailing in C2 lesions, teen groups with mostly C4 lesions, and adults with largely C4–5 lesions [2]. 

SCIs in the neonatal population are quite rare, estimated at 1 case per 29,000 [22,23]. Studies have revealed that birth-related SCIs occur in 10% of perinatal neonatal deaths or stillbirths at autopsy [22].

One major factor in children that can directly influence the risk of cervical SCIs with blunt trauma is the increased relative head size. Compared to adults, children have a larger head-to-body ratio. As a result of a disproportionate head size, and a developing musculoskeletal system, the cervical spine becomes more susceptible to injury from blunt trauma. Compared to adults, children also have weaker muscles in the neck, thus further compromising the stability of the cervical spine. Due to hyperflexibility in the cervical spine and a disproportionately large head size in children, pediatric spines are at risk for the “fulcrum effect”. This condition describes the increased risk of cervical injury as the C2/C3 spine becomes a pivot point for cervical motion, thus increasing the likelihood of cervical injury [24]. Nonetheless, as the spine matures and head size decreases in relation to the rest of the body, this pivot point is shifted caudally as children age, making cervical injuries less likely [25].

### 4.2. Etiology

The etiology of the pediatric SCI differs from that of the adult SCI. In adults, the most common cause of SCIs is traumatic injuries, while in children, non-traumatic causes are more common [26]. The most common non-traumatic causes of pediatric SCIs include congenital anomalies, spinal cord tumors, infections, and vascular malformations [27]. Alternatively, traumatic causes of pediatric SCIs include falls, sports-related injuries, motor vehicle accidents, and child abuse [28]. Overall, there are generally two main causes of SCIs in young children and adolescents. A retrospective study of consecutive SCIs at a single pediatric trauma center reported 52% of injuries to be caused by motor vehicle accidents and 27% caused by sports injuries [29]. Furthermore, most studies reporting SCIs in pediatric populations indicate a greater prevalence of SCIs in males than females [21].

It is important to highlight that neonatal etiologies differ from the rest of the pediatric population and occur mainly during delivery secondary to excessive extraction, rotation, or hyperextension of the neck [30].

### 4.3. Pathophysiology

Spinal cord injuries can be broken down into traumatic and non-traumatic forms of insult. Traumatic injuries result from a direct and abrupt mechanical insult to the spinal cord due to the disruption of the vertebral column (Figure 1) [31]. These kinds of injuries are often accompanied by permanent motor deficits, ranging from mild limb numbness and weakness to complete paralysis, as well as sensory and autonomic impairments. Alternatively, non-traumatic SCI refers to an injury to the spinal cord that is non-mechanical and rather the result of a gradual damage to the spinal cord over time. Despite the differences in etiology, both traumatic and non-traumatic SCIs share various pathophysiological features, including apoptosis, inflammation, axonal degeneration, and changes to vascular permeability, as demonstrated in animal models of both diseases [32].

#### 4.3.1. Traumatic SCI

##### Primary Injury

Traumatic SCIs can be further broken down into primary and secondary phases of injury. The primary phase refers to the initial mechanical damage that directly impacts the spinal cord. This involves the physical disturbance and structural damage to the spine, including bone fracturing and tearing of the spinal ligaments [33]. The primary injury itself can be categorized into two main types: contusion and compression injuries. Contusion injuries are characterized by the direct impact or compression of the spinal cord, leading to localized hemorrhaging/bruising near the injury site. On the other hand, compression injuries occur when the spinal cord is indirectly compressed, for example, by a bone structure or blood from a nearby hematoma. Thus, compression injuries often result in tissue deformation, vascular damage, and the disruption of axons.

##### Secondary Injury

The secondary phase of injury is triggered by the primary phase and involves a series of molecular and cellular events that further exacerbate the chemical and mechanical damage to the spinal tissue. These events begin from hours to days after the initial injury and ultimately lead to neuronal death and cellular dysfunction. Temporally, secondary injury can be subdivided into acute (0–48 h post-injury), subacute (2–4 days post injury), and intermediate/chronic (2-weeks-to-6-months post-injury) phases [32].

The acute-to-subacute period occurs immediately after the initial injury and continues to last up to 4 days post-injury. Based on evidence from animal models, the primary insult triggers a significant inflammatory response, characterized by the activation of immune cells and release of pro-inflammatory cytokines and chemokines. This proinflammatory state is necessary for the phagocytosis of myelin debris; however, it can contribute to additional tissue damage due to the production of free radicals as by-products of cytokine release [32]. Reactive oxygen species (ROS) ultimately lead to the oxidation of lipids, proteins, and DNA within the spinal cord tissue, causing further necrosis and apoptosis and worsening the state of the injury microenvironment [34]. Furthermore, glutamate exotoxicity occurs, characterized by the release of high levels of glutamate from apoptotic neurons and glia, leading to the overactivation of glutamate receptors, an influx of calcium ions, and excitotoxic neuronal death. Ultimately, these cellular processes of excitotoxic cell death and inflammation promote one another, perpetuating the secondary damage [35]. In addition to this, a traumatic SCI inevitably causes damage to the spinal cord vasculature, leading to ischemia that can persist up to weeks after the initial insult. Long-term ischemia plays a role in additional neuronal and glial cell death and spreads from the injury site. Importantly, the acute phase sets the foundation for subsequent phases and provides a critical time window for interventions to potentially limit further secondary damage.

As the subacute secondary injury persists, it leads to the chronic secondary phase of SCI, characterized by the formation of cystic cavities, maturation of the glial scar, and axonal dieback [33]. Beginning in the acute phase of injury, the formation of cystic cavities occurs, where extensive cell death leads to the formation of fluid-filled cysts containing macrophages and connective tissue. Eventually, these individual cystic cavities merge, forming a barrier that inhibits axonal regrowth and cell migration [36]. Around the cystic cavities, a perilesional zone forms, where reactive astrocytes proliferate and interweave with one another, forming an inhibitory structure known as the glial scar. Aside from astrocytes, the scar is formed by a combination of various ECM proteins, for example, chondroitin sulfate proteoglycans (CSPGs) and NG2 proteoglycan. Along with astrocytes, these proteins form the glial scar, which inhibits axon regeneration by impeding neurite outgrowth [37]. Importantly, the glial scar is believed to play a dual role, as it is beneficial in isolating the injury site to help limit the spread of cytotoxic molecules and inflammatory cells; however, the role of the glial scar is still being investigated [38].

#### 4.3.2. Non-Traumatic SCI

Alternatively, pediatric SCIs may result from not only mechanical causes such as the ones mentioned above but also non-mechanical mechanisms. For example, congenital anomalies can lead to the abnormal development of the spinal cord, which can result in an SCI without any mechanical injury. The most common non-traumatic causes of SCIs include neoplasms or spinal cord tumors that can be either benign or malignant and lead to compression of the spinal cord, resulting in an SCI without direct trauma. Aside from tumors, transverse myelitis is another common cause of non-traumatic SCI in children, involving the inflammation of both sides of the spinal cord, often leading to damage to myelin and resulting in impairments in nerve signals [5]. Compared to adults, neurological recovery following an SCI is reported to be better in pediatric populations. However, the evidence is very weak for this, as no studies have compared adults and children directly in terms of recovery following an SCI. 

#### 4.3.3. Developmental Variants and Anomalies

Os odontoideum—This is an anomaly characterized by a well-corticated ossific density along the superior margin of a relatively hypoplastic or foreshortened base of the dens. There is debate surrounding its etiology and whether it is congenital or traumatic and if it happens earlier in the development of the spine [39].Persistent ossiculum terminale—This refers to a failure of fusion of the secondary ossification center along the superior margin of the dens. The ossiculum terminale is supposed to ossify during the mid to latter half of the first decade of life and fuses during the second decade of life [7,40]Odontoid synchondrosis slip—As known, there are five ossification centers for the axis present at birth and two for the odontoid. In between these centers, there is a junction that has been shown to have synchondrosis characteristics. It lies at a level below the C1–2 articulation and is weaker than the ligaments or bone, making it the least able to resist force [14,41]. With trauma, this structure can be avulsed or slip, even if there are no bony fractures [13].

## 5. Pediatric SCI Diagnosis

### 5.1. X-ray, CT(A), and MRI

Although the identification of spinal injuries is of huge importance, it is always important to remember that imaging in the pediatric population has some particularities and concerns due to the increased risk of radiation in these patients [1].

There is little evidence regarding thoracolumbar injury imaging, but for cervical injuries, there is a large volume of works in the literature [42]. The mechanism of injury should guide decision-making regarding which imaging modality should be considered, but more recently, the Delphi study has been arguing against its role [12]. It is important to remember, though, that specific mechanisms of injury and the presence of symptoms can guide the path to follow for diving injuries, seatbelt-type injuries, non-accidental trauma (NAT), and the presence of back or neck pain, amongst others. It is also important to mention that the identification of an SCI at any level indicates the need to perform a full spinal column evaluation in terms of imaging, as there is evidence to support the possibility of noncontiguous injuries up to 6% and contiguous in 32% of patients [43,44].

The National Emergency X-Radiography Utilization Study (NEXUS) is one of the most used methods to clear cervical pediatric SCIs due to its validation in a study performed by Viccelio and colleagues [45]. NEXUS utilizes five criteria: (i) presence of neurologic deficits, (ii) midline spinal tenderness, (iii) altered level of consciousness, (iv) intoxication, and (v) distracting injuries to define which is the best imaging option [46]. It is recommended to perform anteroposterior and lateral cervical spine X-rays or high-resolution CT scanning in children younger than 9 years with a history of trauma and who are not alert or not conversant; have a neurological deficit, neck pain, or a painful distracting injury; are intoxicated; or have unexplained hypotension [47]. In the population aged 9 years and older, it is also recommended to add the open-mouth cervical spine X-ray (1a). Additionally, evidence supports cervical spine screening in children after an SCI who are not communicant due to age (<3 years old) and were involved in a motor vehicle collision or a fall from a height > 10 feet, have suspected NAT mechanisms, or have a GCS < 14 [42].

Dynamic studies, such as flexion and extension fluoroscopy or radiographs, should be considered when the child’s clinic or static X-rays suggest cervical spinal instability [48,49]. The use of CT is not known to clear all cervical spines in the pediatric population, and, thus, it should be employed with caution due to the amount of radiation in this test, unless there is the risk of a potential atlanto-occipital dislocation (AOD) [50,51]. Magnetic Resonance Imaging (MRI) is an important information provider about ligamentous injuries that may influence surgical management; however, it requires cooperation or sedation in cases where the child cannot tolerate it. This latter imaging study may also provide prognostic information regarding existing neurological deficits [42,52]. In MRIs, T1-weighted images help with the evaluation of cord morphology, as well as anatomy, and a T2 low signal shows acute hematomas [53]. Subacute hematomas appear hyperintense on both T1- and T2-weighted imaging. Lastly, chronic hematomas appear hypointense on T2-weighted imaging. Edema appears hyperintense in comparison with healthy normal nervous tissue in T2. This is important since these patients with hemorrhaging tend to have worse outcomes, while patients with single-level edemas have better outcomes [6,54].

The American Association of Neurological Surgeons (AANS)/Congress of Neurological Surgeons (CNS) Joint Guidelines Committee recommendations based on level 1 evidence suggest the use of computerized tomography (CT) in suspected atlanto-occipital dislocation to determine the condyle-C1 interval (CCI) [42,47]. It is also relevant to highlight that CT cervical may be the chosen modality in patients who are obtunded, have polytrauma, or had a high-risk mechanism of injury [54]. Level 2 evidence supports the recommendation against performing imaging in children > 3 years of age without the criteria proposed above [42,46], with motor-vehicle-accident trauma mechanisms, falls from heights greater than 10 feet, and nonaccidental trauma [55]. In cases that do not meet these criteria, cervical spine X-rays or CT scans must be performed.

It is also important to state that CT-scan radiation is an important topic to address regarding the pediatric population, since the pediatric population is more at risk for malignancies than adults [56,57]. Due to that risk, the new trends are to have reduced radiation CT scans or even evaluating its real need. This is a very subtle call to be made, since there is a risk of missing diagnosis because of suboptimal image quality as a consequence of exposure settings being too low [58,59], so the discussion should always center on minimizing the risks for the patients.

### 5.2. Contrast to Adults

#### 5.2.1. Upper Cervical Spine Injuries

##### Craniocervical Junction Injuries

Due to the anatomical differences mentioned earlier, atlanto-occipital distraction injuries are the most common cervical spine injuries in the pediatric population [60]. On radiographs, the most sensitive measurements for these injuries are the basion-dens interval and C1/2:C2/3 interspinous ratio [61]. Despite the disruptions being incomplete, there is often minimal-to-no displacement in the craniocervical junction with these injuries. In these cases, the measurements mentioned may not be sensitive; however, soft-tissue findings such as retroclival hematoma, fat stranding between the basion and dens, extrathecal collections, and hemorrhaging can indicate an underlying distraction injury [54]. Lastly, as occipital-condyle fractures can be more common in children than in adults [62], it is important to identify avulsion fractures on this structure, as they can be associated with atlanto-occipital dissociation [63].

##### Atlas Injuries

Atlas injuries are rarely seen in young children; these types of injuries are more frequent in adolescents [54,64]. C1 fractures can be missed on radiographs, but the displacement of the lateral aspect of the lateral mass of C1 relative to the lateral borders of C2 from an odontoid view can be an indication of the lesion. It is also important to highlight that unfused C1 synchondrosis can be mistaken for a fracture. In these cases, asymmetric widening of the synchondrosis or adjacent soft-tissue edema suggests an injury in the region, which is better identified through MRI.

Lastly, rare avulsion of the C1 anterior arch, seen almost exclusively in the pediatric population, is one lesion to be aware of in cases where hyperextension is part of the injury mechanism [64].

##### Atlanto-Axial Injuries

Less often than the craniocervical distractions injuries, atlanto-axial injuries usually occur as partial rather than complete lesions [62]. Patients with underlying disorders that create ligament laxity, such as Down, Morquio, and Marfan syndromes, are at a higher risk of presenting with these [65]. Suggestive radiological findings in these cases are a widened atlantodental interval and disruption of the spinolaminar line, and in cases that the subluxation of the atlas relative to the axis occurs, spinal-canal narrowing can be seen [62]. It is important to highlight the possible occurrence of Grisel syndrome in the pediatric population [66]. It consists of a non-traumatic subluxation of the atlanto-axial joint that is associated with retropharyngeal abscess, otitis media, and upper-respiratory-tract infection [66].

Furthermore, young children can present with instability between C1 and C2, causing an atlanto-axial rotatory subluxation due to ligamentous and capsular hypermobility [67]. There are four types of these injuries: Type 1 occurs when there is an abnormal rotation to the atlas relative to the axis without subluxation across the C1–C2 level. Type 2 injuries are rotational deformities where there is mild anterior subluxation of C1–C2, corresponding to a widening of the atlantodental interval to 3–5 mm. In type 3 rotational injuries, the interval is widened to more than 5 mm. Finally, in type 4 rotational injuries, there are deformities with retrolisthesis at C1–C2, resulting in displacement of the C1 anterior arch to the dens. The rotational component of these injuries is demonstrated as the medial offset of the C1 lateral mass, which is rotated anteriorly, and lateral offset of the opposite lateral mass upon open-mouth view [68].

##### Axis Injuries

In the pediatric population, odontoid fractures (type 1 and type 3 dens fractures) occur frequently [62]. Type 1 and type 3 occur through the apicodental and subdental synchondroses, respectively. Type 2 odontoid fractures become more prevalent in adolescents after the fusion of the synchondroses, similar to the injuries seen in adults [62].

It is important to acknowledge os odontoideum, a rare anatomical variant where the odontoid ossification center fails to fuse with the C2 vertebral body; it can be distinguished from a dens fracture by the presence of smooth, corticated margins between the ossicle and C2 vertebral body. While os odontoideum is not a fracture, it is important to identify and report since it can lead to instability, pain, myelopathy, and vascular issues [69].

The C2 vertebra is also susceptible to traumatic spondylosis, often called a “hangman’s fracture”, which refers to a fracture between the pedicle and inferior articular facet. Traumatic spondylolysis is rare in children, especially in those under the age of 9 years. Pseudosubluxation of the cervical spine is more common in the pediatric population, particularly in those under 8 years of age [70]. It is characterized by an apparent misalignment of the cervical spine, but it is actually a normal finding in this age group, with an offset of greater than 3 mm in the anterior and/or posterior longitudinal line while maintaining normal alignment of the spinolaminar line at the C2–C3 level [64].

#### 5.2.2. Subaxial Cervical Spine Injuries

Injuries below the C2 level can be classified using the AO Spine subaxial CSI classification system [71], which considers four criteria: injury morphology, facet injury, neurologic status, and case-specific modifiers. The morphology is divided into three categories: type A (compression fractures), type B (disruption of tension band without translation or dislocation), and type C (displacement or translation of vertebral bodies) [71]. AO Spine A injuries are more common in patients older than 8 years, while AO Spine B and C injuries occur more frequently in children [62]. Soft-tissue changes may be the only evidence of subaxial injury, and close inspection of cervical soft tissues is crucial. Physiological vertebral body wedging and unfused epiphyseal-ring ossifications are common mimics of fractures in the subaxial cervical spine [72,73].

It is important to note that most vertebral body wedging in patients under 8 years is physiologic and not indicative of fractures. As the epiphyseal-ring cartilage begins to ossify, it can resemble a small avulsion fracture from hyperextension injury; however, this injury is rare in pediatric patients. Small calcifications along the corners of the vertebral body are likely to be normal physiologic mineralization in the absence of malalignment- or soft-tissue findings [72].

### 5.3. SCIWORA 

A spinal cord injury without radiographic abnormality (SCIWORA) remains an entity of concern in pediatric patients. Recent findings show that the more widespread usage of MRI has improved the understanding regarding this specific pattern of injury in the pediatric population [74].

White matter integrity can be assessed by diffusion-weighted MRI (DWI), as shown by Shen et al. [75], where even with T1- and T2-weighted normal images, the DWI in SCI patients exhibited an abnormal signal [75]. For structural-damage evaluation, diffusion tensor imaging (DTI) is another tool that can be used, as it can examine and differentiate white matter from gray matter. Mulcahey et al. found a better association between DTI and the examination by using the International Standards for Neurological Classification of SCI (ISNCSCI); however, it has been validated only for adults [75].

MRI findings in children with SCIWORA range from normal to complete cord disruption, along with the injury of ligaments and discs [76,77,78]. Therefore, level III evidence [79] suggests that magnetic resonance should be performed in children with clinical spinal symptoms or suspected injury, in addition to radiographic screening of the entire spinal column. In these patients, flexion–extension radiographs are recommended in the acute setting and in late follow-up, even if the MRI shows no extra neural injury. There are no recommendations regarding angiography or myelography in children with SCIWORA [79].

Lastly, there are some recommendations to perform a somatosensory evoked potential (SSEP) to screen this group of patients, especially in cases where there is subtle posterior-column dysfunction and the clinical findings are not conclusive [78]. SSEPs also help to distinguish intracranial, spinal, and peripheral nerve injuries in head-injury, comatose, or pharmacologically paralyzed patients [78].

## 6. Pediatric SCI Classification

As discussed above, pediatric SCIs have unique characteristics that differentiate them from SCIs in adults, including differences in etiology, spinal cord anatomy, and physiological responses to injury. As such, classification systems for pediatric SCIs have been developed to aid in diagnosis, prognosis, and treatment planning. This section reviews the current classification systems for pediatric SCI. 

The most widely used classification system for SCIs is the American Spinal Injury Association (ASIA) impairment scale, which classifies SCIs based on the neurological level of injury (NLI) and the completeness of injury [80]. This set of standards is known as the International Standards for Neurological Classification of SCIs [80]. The NLI is determined by the lowest spinal segment with normal sensory and motor function, while the completeness of injury is either classified as complete or incomplete. Complete injuries characterize injuries where there is an absence of motor and sensory function below the NLI, while incomplete injuries involve some motor and/or sensory function that is maintained below the NLI. 

Currently, the ASIA classification system for SCIs has been validated only in adults and not in pediatric populations. Importantly, the ISNCSCI is not considered appropriate for use in children younger than 6 years old [81]. To better apply the scale to pediatric populations, modifications must be made to account for differences in spinal cord anatomy and development. In its current state, the ISNCSCI may not be able to accurately measure the neurological consequence of SCIs in young children, and only an estimated neurological level can be provided. 

Developing tools for assessing neurological impairment in infants is important for improved diagnosis. For example, infant-specific motor scales may be beneficial to assess motor impairment more accurately in infants through the monitoring of physiological variables, including heart rate and blood pressure, during sensory testing. Alternatively, establishing correlations between imaging studies in children, such as CAT scans, MRI, or DTI, and the ISNCSCI motor and sensory scores may help establish a standardized assessment approach for young children, who lack the cognitive ability to participate in the assessments [82]. 

In addition to impairment-based classification systems, functional classification systems have been developed to assess the impact of an SCI on a patient’s daily life. One example of this is the Spinal Cord Independence Measure (SCIM-III), which assesses an individual’s ability to perform activities of daily living, such as grooming, dressing, as well as bladder and bowel management. The SCIM-III is especially relevant for the pediatric SCI population, as pediatric rehabilitation primarily focuses on restoring daily function and independence skills [82]. The assessment also includes a self-report version, which provides a substitute to the performance-based version primarily in community-based settings or for long-term monitoring. 

Importantly, classification systems for SCIs should consider the etiology, as well as the level and extent of injury, which is important in the prognosis and treatment of these patients. The etiology—for example, traumatic versus nontraumatic causes of SCI—can have serious implications for the interventions required for treatment. Furthermore, the level of injury and region of spinal cord tissue affected are important for understanding its effects on motor and sensory function, as well as in guiding treatment planning. Ultimately, the consideration of function, etiology, and the type of injury is important in classification systems for pediatric SCIs. Current classification systems are continually evolving, and further research is needed to improve their accuracy and utility in the diagnosis, prognosis, and treatment of pediatric SCIs. Recently, an instrument for spinal cord injuries’ inventory (SC-SCII) and self-care self-efficacy scale (SCSES-SCI) was developed in the adult population [83]. It has been shown to be a valid and reliable instrument for assessing self-care in these patients, and it can be further tested and validated in younger patients.

## 7. Pediatric SCI Management

Fortunately, pediatric SCIs remain a rare entity. However, given the rarity of occurrence, evidence surrounding their management is also limited, and much of the management is drawn from the adult population. The initial management follows the ATLS guidelines, which include the management of airway, breathing, and circulation. There are marked differences in the physiology of pediatric and adult populations that must be taken into consideration in the assessment of pediatric trauma. Both the acute and ICU management of children differ from the equivalent in adults due to the increased importance of airway control, given that the pediatric population is less tolerant of apnea prior to cardiac arrest [3]. Additionally, it is prudent to maintain euthermia in pediatric populations. SCIs can result in hypothermia; this is of greater consequence in pediatric patients given the increased metabolism and resultant O2 consumption in this population, which has downstream implications for coagulopathy [3,84]. Therefore, due to the metabolic differences between adults and pediatric patients, euthermia is essential in preventing further complications of SCI.

The Congress of Neurological Surgeons (CNS) published guidelines on the management of pediatric SCIs, and the suggestions include closed reduction and halo immobilization of injuries to C2 synchondrosis in children. Immobilization is imperative in preventing further SCIs and resultant neurologic deficits.

Implementing conservative management strategies is often preferable in the pediatric population, given that pediatric patients have anticipated growth and anatomical changes that can be impeded by surgical fusion. When dealing with hyperextension injury, it is suggested that conservative management be opted for unless there is ongoing compression, edema, progressive neurologic deterioration, or increased intramedullary pressure, in which case, urgent decompression is suggested. Conservative treatment is the main treatment for children with pediatric acute hyperextension spinal cord injury (PAHSCI).

Surgical fusion is recommended for both spinal instability (irreducible AARF, ligamentous injuries, and irreducible fractures resulting in deformity) and SCIs resulting from canal compression. For SCIWORA, the recommendation is external immobilization for 12 weeks; however, this was not seen to make a difference, as demonstrated by Bosch et al. in 2002. Most often, pediatric SCIs are evaluated for surgical treatment on a case-by-case basis since consideration must be given to the implications on future growth after surgery.

In the neonatal population, surgery is particularly challenging due to the unique anatomy of such patients, the difficulty of achieving adequate internal fixation given bone immaturity, and the potential for growth. Cervical spine injuries in neonates are yet to be better studied, and algorithms, as well as guidelines, are not standardized, but there was a recent report pointing to the possibility of using custom orthoses to treat these patients with a conservative approach [30].

### 7.1. Steroidal Therapy

The role of steroidal therapy in children is less clear given that children under the age of 13 were excluded from the NASCIS trials [85], and additional studies have been limited in their sample size to draw conclusions. However, Zeng et al. suggest that for pediatric acute hyperextension SCIs, steroids can be considered within 8 h of injury. On the other hand, Pettiford et al. explored the insufficient evidence of steroids, calling for increased randomized controlled trials in children to reach a solution; their study alluded to the increased risk of infection without neurologic improvement. It has also been advised that high doses of steroids be reserved for the perioperative period given the risks of wound infection, PE, and death [86,87]. In adults, a 24 h infusion of a high dose of methylprednisolone sodium succinate (MPSS) is suggested as a treatment option within 8 h of the acute event [1]. One can extrapolate this knowledge to older children given the paucity of specific works in the pediatric literature.

Regarding high-dose methylprednisolone pulse, Zeng et al. [88] considered this treatment as an experimental option for PAHSCI in adults, considering children have a better potential for neurological recovery. Furthermore, there is no reliable pulse therapy for children available at present, so it may be used according to the experience. A small pediatric study was published in 2011 that showed a total recovery in 13 out of 25 children between 8 and 16 years within 24 h after the administration of methylprednisolone; nevertheless, the injuries of these patients were limited to spinal cord contusions, and none of them needed surgery [89].

### 7.2. Cardiovascular Management

Patients with SCIs can experience varying degrees of shock, and it is imperative to differentiate between neurogenic shock (NS) and hemorrhagic shock to provide appropriate management [88,90].

Neurogenic shock is distributive in nature, causing hypotension without an accompanying increase in heart rate; thus, patients may respond to intravenous fluid administration but will often require vasoactive support [91]. In such cases, pharmacological support involves the use of α agonists to address hypotension and β agonists to manage bradycardia [88,90].

It is suggested that a MAP > 85 mm Hg be maintained for 7 days post-injury to prevent further ischemia in adults with SCIs. We perceive this knowledge to be potentially applicable to the older pediatric population, i.e., adolescents [88,92]. To date, a paucity of data exists to substantiate the establishment of cutoff points for blood pressure in the pediatric population, and we suggest maintaining age-appropriate blood pressure.

Finally, Parent et al., through a systematic review, concluded that there is no evidence to support neuroprotective strategies such as hypothermia in the pediatric population [21]. Overall, the timing of surgery, ICU, and pharmacological management of pediatric SCIs present a paucity in the literature [93]. However, the works in the literature regarding post-injury complication management, care, and rehabilitation are abundant.

### 7.3. Surgical Timing

Although the literature is still limited within the pediatric population, a recent update in the guidelines for adults was compiled by experts and is now in press. Previous 2017 guidelines formulated a weak suggestion for early surgery, based on a systematic review, using 24 h as the threshold between early and late decompression [94]

After this first guideline, several studies assessed this recommendation, with new evidence emerging to change the strength of the recommendation to moderate. Early surgery (≤24 h after injury) should be offered as the primary approach for adults with an acute SCI regardless of the level of injury, as recovery in a ≥2 grade ASIA impairment Score (AIS) was more likely to occur within 6 and 12 months [95,96,97,98] when patients were decompressed within 24 h compared to after 24 h.

At this point, it is not possible to determine the effectiveness of early surgery in different subpopulations, especially in the pediatric group, but these recommendations can guide surgical decisions. It is also important to point out that further studies on ultra-early surgery must be performed to determine the impact on neurological recovery. Additionally, there is a lack of definition of what constitutes effective spinal cord decompression to individualize care, especially in the pediatric population, although most injuries do not result in the compression of neural elements.

## 8. Complication Management

Although SCI complications in children are similar to those found in adult patients, the pediatric population has a few particularities to be taken into consideration [1], such as difficulty expressing feelings, which can lead to a delay in diagnosing complications. It is very important to actively monitor for spinal deformity, hip dislocation, hypercalcemia, pain, venous thromboembolism, and autonomic dysreflexia. Bowel and bladder function are also usually impacted in SCI patients and must be managed carefully.

### 8.1. VTE Prophylaxis

Venous thromboembolism (VTE) is a well-known condition found in hospitalized patients with limited mobility, especially in the adult population. Under the age of 15, VTE is an occasional and rare event, with fewer than 5 cases per 100,000 [52,99], and this number tends to rise at an exponential rate beyond that [99]. This risk can be elevated in polytraumatized patients and in cases with traumatic-brain-injury association [100,101]. In these cases, the use of central venous catheters can increase the risk of VTE as well [100,101]. In pediatric traumatic spine injuries, the incidence of VTE also occurs in a low percentage of patients, and the risk is increased depending on the severity of the trauma [102], as well as concomitant injuries such as SCI, cranial hematoma, and lower-extremity injuries [101].

The consensus for VTE chemoprophylaxis is limited for post-puberty children, as in the adult population, for 8 weeks [103]. In prepubescent children, there is little evidence for the recommendations, and treatment is usually restricted to mechanical prophylaxis, such as the use of pneumatic compression devices, and graduated compression stockings [103] if appropriate sizing can be achieved. If using commercially available stockings is not possible for smaller children, for example, custom-made lower-extremity stockings should be taken into consideration. Elastic wraps are discouraged in these patients due to the risk of venous obstruction and compartment syndrome, amongst others [104].

In cases where the children meet the criteria for chemoprophylaxis, it should be started soon after the injury [103] if there is no active bleeding or low risk for it. Enoxaparin dosage in patients younger than 2 months is 0.75 mg/kg every twelve hours, and it is 0.5 mg/kg twice a day or 1 mg/kg once a day in those older than 2 months of age [105].

### 8.2. Autonomic Dysreflexia (AD)

The development of autonomic dysreflexia usually takes place in T6 or higher injuries, as reported by Shcottler et al. [106], in approximately 51% of children, and it can be life threatening [107]. As seen in the adult population, children with SCIs show lower baseline blood pressure [53]. The diagnosis for autonomic dysreflexia is made if the baseline blood pressure is exceeded by more than 20 mmHg [103]. Other symptoms described are sudden-onset severe headaches, with flushing and sweating above injury level and cool skin with piloerection below [108].

It is also important to consider this diagnosis in younger children that are unable to communicate or are too young to express themselves if they are presenting with unusual drowsiness or with a new onset of irritability [109].

Treatment involves conservative measures by removing all potential irritants. The popular pneumonic to remember is “6 Bs”: bladder (urinary retention or infection, nephrolithiasis, and blocked catheters), bowel (impaction and constipation), back passage (hemorrhoids and fissures), boils (skin damage), bones (fractures), and babies (pregnancy and sexual intercourse) [110].

Pharmacologic measures include fast-acting anti-hypertensives such as nifedipine, and for the recurrent cases, prazosin or terazosin [6]. For blood-pressure control, it is also important to position the patient upright to promote orthostatic drop and loosen any constrictors, such as tight clothing or dressing [103].

Correct management is extremely important since AD can lead to seizures, stroke, and intracerebral hemorrhage [111].

### 8.3. Pain and Spasticity

Pain management in patients with SCIs can be a challenging matter, and it is best to have an experienced pain-management team in these cases [112]. Although it is always better to start with medications that do not impair respiratory drive and to avoid the ones that can cause cardiac and behavioral complications, many patients will need medications targeting neuropathic pain [113]. Gabapentinoids, selective serotonin reuptake inhibitors, tricyclic antidepressants, and lidocaine patches should be considered if the patient’s condition allows their use.

Spasticity is another important feature of SCI patients that will only appear after the spinal cord shock period, which is highly variable and can be delayed to up to 2 months after the injury [114]. Managing spasticity must be individualized since some degree of it can be tolerable. The first step is to exclude potential nociceptive factors that can exacerbate spasticity, while physical therapy must be introduced before considering oral medications [114,115,116]. Afterwards, if the patient is still presenting with generalized spasticity, an oral medication should be started. The most commonly used drugs include baclofen, diazepam, dantrolene sodium, tizanidine, and clonidine [114], and the choice of which one to start with is based on clinical experience, commencing at a low dose and titration to optimal dose [117]. Usually, baclofen is the drug of choice, with diazepam often added as an adjunct initially at night [118]. Clonidine and tizanidine may also be added to diazepam, with one single nighttime dose at first [118]. In supraspinal injuries, dantrolene may be preferred to avoid sedation from the cited drugs [117]. Focal spasticity, on the other hand, is better managed with intramuscular botulinum toxin (BTX) injections and phenol/alcohol neurolysis. Electrical stimulation or ultrasonographic guidance may be used for better muscle localization [114].

Patients with significant functional managements that are refractory to the medications should be treated with surgical management. Intrathecal baclofen (ITB) is highly effective and is delivered in the intrathecal space via a catheter connected to an implanted pump in the abdomen. This method allows for a minimized depressive nervous system effect; however, there is always the risk of potential complications such as infections, cerebrospinal fluid leaks, and problems with the catheter [119].

### 8.4. Bowel and Bladder

Although it may not be present right after the SCI, children can have bowel and bladder dysfunction. Depending on the level of injury, patients will have either upper or lower motor neuron bowel patterns. An upper motor neuron bowel pattern results in hyperreflexic bowels and a spastic anal sphincter, and the patient will present with constipation and retention [112]. With a lower motor neuron bowel pattern, the patient will present with an areflexic bowel and atonic external anal sphincter, which will lead to a mix of constipation and incontinence [112].

The goal for these patients is to achieve continence, but, more importantly, patients should be started on a bowel regimen for regularly timed bowel movements. In the upper motor neuron presentation, the program should include digital stimulation triggering retrocolic reflux, along with oral and rectal medications, if needed [112,114]. It can also be helpful to have gravity-aided postures and suprapubic pressure (Credé maneuver) [114,120]. Retrograde irrigation systems such as enema and inflatable rectal catheters can be interesting additional strategies, as well [121]. In lower motor neuron programs, manual evacuation may be necessary [122]. Ultimately, for children with significant difficulty in achieving continence, it may be beneficial to perform the anterograde continence enema procedure [122].

Neurogenic bladder is another important concern in these patients that need managing. The primary goals are to preserve renal function and to promote continence [120,123]. The standard of care recommended by the Consortium of Spinal Cord Medicine [103] is intermittent urinary self-catheterization. Younger pediatric patients cannot perform the procedure independently; however, independence should be stimulated as soon as possible.

Patients presenting with a spastic detrusor muscle may benefit from anticholinergic drugs or detrusor BTX injections. If conservative methods are not able to help with satisfactory continence, surgical intervention may be considered [124].

### 8.5. Spinal Deformity and Hip Dislocation

One other important feature in children with SCIs is spinal and hip deformities. Spinal deformities are common, and scoliosis can develop in up to 97% of these patients before a growth spurt [114]. This is the cutoff for the pediatric population, and children injured before the growth spurt should be monitored closely. Usually, it can be treated with bracings, but if it fails and the Cobb angle is >40°, surgical correction is indicated. Surgery can also be indicated in patients greater than 10 years old with a rapid deformity progression and functional problems or pain.

Hip dislocations and subluxation are also common in children under the age of 10 years old, occurring in >90% according to McCarthy et al. [125]. Positioning and surveillance are very important in these patients, as surgical intervention may be needed [114].

### 8.6. Hypercalcemia

Around 23% of children, especially adolescent males, can be affected by hypercalcemia [126]. The classical symptoms are abdominal pain, polyuria, vomiting, generalized malaise, and psychosis. An acute abdomen is one misdiagnosis that can be made in these patients, leading to unnecessary surgeries.

Hypercalcemia can also increase the risk of nephrocalcinosis, urolithiasis, and ultimately renal failure [126]. Treatment generally consists of aggressive hydration with furosemide diuresis and bisphosphonates in some cases [114,126].

## 9. Pediatric SCI Recovery

### 9.1. Rehabilitation

Although a pediatric SCI is a devastating condition that can cause lifelong neurological sequela impacting the individual and their family, the research on rehabilitation in this population is still scarce, focusing mainly on the adult group [127]. Recently, a project named The Spinal Cord Injury Rehabilitation Evidence (SCIRE) commenced a combined effort to produce chapters on relevant topics related to SCIs in the pediatric population [127], but there is still a lack of research on this area to fill in the gaps. It is important to acknowledge that the rehabilitation of these patients is complex and involves their mental, physical, and social health.

Mental illness, especially depression, is common in individuals with an SCI. It is estimated that 22% of the adult population presents with depression [50] but in the pediatric population, the prevalence seems to be lower [127,128], with a certain predominance in older adolescents (12–18 years). Anxiety is more common in older female adolescents, and clinical factors such as a short duration of injury are more associated with mental-illness development [129].

Physically, these patients are impaired in many ways, and proper management is required regarding pain, venous thromboembolism, genitourinary issues, and deformities, as stated previously. Nevertheless, it is also important to focus on cardiovascular care and skin health. Although evidence regarding cardiovascular care is very limited in pediatric medicine [127], relevant studies targeted general outcomes, including body composition (i.e., total lean mass, fat mass, % body fat, and bone mineral content/density), anthropometric measures (i.e., weight, height, and body mass index), metabolic efficiency (e.g., fasting lipids/glucose and resting metabolism rate), cardio-respiratory function (e.g., heart rate), and functional performance (e.g., muscle strength and power input) [127]. Overall, the pediatric population with SCIs exhibits body composition changes such as higher fat mass, lower total lean mass, and increased body mass index, amongst other changes that can predispose an individual to poor cardiovascular health [130,131,132,133,134,135]. An SCI can also predispose individuals to pressure-induced skin injuries due to the impaired autonomic regulation of subcutaneous blood flow and skin moisture levels, reduced skin temperature reactivity, decreased immune response, and changes in connective tissue composition [136]. Aside from pressure injuries, it is also possible that other skin comorbidities arise in these patients, such as self-inflicted wounds [137,138,139] and latex allergy [140].

In patients with mobility impairments leading to the inability to sit upright, stand, transition from sitting to standing, and/or walk, a historical compensatory orthopedic approach is the solution to achieve alternative ways to function. It includes braces (e.g., parapodium and knee–ankle–foot orthoses) [141,142], assistive devices [143], or electrical stimulation [144,145,146,147,148,149] alone or combined with braces.

Ultimately, the major life areas are on the scope of care for these patients. Returning to school is a primary rehabilitation goal, and physical accessibility and social participation must be reassured. It is important to involve the physical and occupational therapist in this process, as well as use assistive technology and include the child’s perspective on this process [127,150,151]. Community integration is also important, and it may mitigate some of the long-term consequences of SCI. Individuals with pediatric onset SCI are more likely to report greater functional independence, less pain, and fewer comorbidities requiring medical intervention as compared to individuals with an adult-onset SCI [127].

### 9.2. Advances in Rehabilitation Engineering to Facilitate Independence

It is important to emphasize exciting advances within the scope of neurorehabilitation therapies with brain–computer interfaces that allow patients to restore lost function following an SCI. Over the past 20 years, numerous interfaces have emerged and achieved success for upper-limb applications involving grasping and reaching [152,153,154]. However, tetraplegic patients still encounter difficulties, as current implantable neural bypass systems require the patient to be constantly connected to an external power source, thus limiting their usage outside of the laboratory [152,155]. Furthermore, the algorithms used in these cases rely on the activity of a single neuron, a function that can deteriorate over time. Consequently, programs have emerged based on electrical activity acquired through methods such as EEG (electroencephalography) or electrocorticography, among others [156,157,158].

In addition, it is important to mention exoskeletons as an emerging therapy for SCIs. The use of exoskeletons in acute rehabilitation and daily activities offers a new approach to the complex processes of spinal cord regeneration and repair, which has hindered progress in SCI treatment. Exoskeletons address the limitations with body-weight-supported treadmill training, which is often not possible due to muscle atrophy following an SCI [159,160]. Exoskeletons support weakened stabilizer muscles, increasing the patient and therapist’s work efficiency. Ongoing clinical trials with treadmill-based and mobile exoskeletons may revolutionize therapy for chronic SCI patients in the near future, improving their cardiopulmonary function, muscle physiology, and walking performance [160,161,162].

## 10. Experimental Research and Future Treatments

An SCI is a devastating condition that results in the partial or complete loss of motor, sensory, and autonomic functions. Traditional treatment approaches have focused on rehabilitation and symptom management, offering limited prospects for neural recovery. However, in recent years, neuroregenerative therapies have emerged as a promising avenue for promoting neural repair and functional restoration (Table 1).

### 10.1. Neuroprotective Strategies

#### 10.1.1. Pharmacological Therapies

##### Riluzole

Riluzole is a benzothiazole sodium channel blocker that is commonly used in patients with amyotrophic lateral sclerosis (ALS); it acts as a glutamatergic modulator. It has shown favorable results in preclinical trials, promoting recovery in models of traumatic spinal cord injuries (tSCIs) and in early phase clinical trials [163,164,165,166].

In a placebo-controlled study, Meshkini et al. demonstrated better sensory and motor outcomes for patients with AIS grades A–C, with improvement on pain reports after an 8-week-long treatment with riluzole at 50 mg twice a day [165]. In a prospective multicenter phase I matched-comparison group trial, Grossman et al. also demonstrated higher motor scores in riluzole-treated patients than controls at 90 days postop; however, 14–70% of patients had elevated levels of liver enzymes and bilirubin [167]. Caglar et al. demonstrated similar results highlighting the effects of riluzole on SCIs in rats. The results showed that early riluzole infusion causes lower histopathological spinal-cord-tissue damage and improves the number of surviving glial cells and neurons [168]. In a comparison preclinical study performed by Hosier et al. with glibenclamide, hypothermia, riluzole, and control groups, the last two groups had a higher mortality rate (30%) and lower efficacy [169].

A more recent international multicenter, randomized, double-blinded, placebo-controlled phase III trial (ClinicalTrials.gov: NCT01597518) entitled the Safety and Efficacy of Riluzole in Acute Spinal Cord Injury Study (RISCIS) was undertaken [93,163]. Patients with ASIA scale A–C, cervical (C4–C8) tSCI, and <12 h from injury were enrolled, and the group receiving riluzole at an oral dose of 100 mg twice per day for the first 24 h, followed by 50 mg for 13 more days, showed improvement in upper-extremity motor (UEM) scores of 1.76 (95% CI: −2.54–6.06) and in total motor (TOTM) scores of 2.86 (CI: −6.79–12.52) after 180 days. Improvements in the riluzole group were also seen at 6 months compared to the placebo group. It is important to note that the primary analysis did not achieve the predetermined endpoint efficacy for riluzole, likely due to insufficient power. However, preplanned secondary analyses showed significant changes in all subgroups of cervical SCIs (AIS A, B, and C).

##### Granulocyte Colony-Stimulating Factor (G-CSF)

G-CSF is a cytokine glycoprotein that is known for its ability to promote the survival, proliferation, and differentiation of cells of a neutrophil lineage. Research in animal models shows that G-SCF suppresses neuronal and oligodendroglial apoptosis [170], protects myelin, and reduces inflammatory cytokine expression to enhance recovery after an SCI [171,172]. A phase III double-blinded, randomized, placebo-controlled trial performed in Japan, named G-SPIRIT, recently investigated the effects of 400 μg/m^2^/day, administered for five consecutive days intravenously, in patients with acute AIS grade A and B SCIs. The group showed improvement in ASIA motor scores at 6 months (*p* = 0.062) and 1 year (*p* = 0.073) after drug administration in the G-CSF group compared to the placebo group, although it failed to show any significant effect in the primary end point (at 3 months) [173]. A previous phase III double-blind RCT performed by Derakhshanrad et al. also demonstrated significant improvements in the ASIA Scale (AIS) motor scores in patients with a chronic SCI receiving 300 μg/day over the course of one week, with an improvement of 5.5 (±0.62) points versus 0.77 (±0.20) in the placebo group after 6 months of G-CSF administration [174].

#### 10.1.2. Nonpharmacological Neuroprotective Strategies

##### Systemic Hypothermia

Systemic hypothermia serves as a therapeutic intervention that aims to reduce the basal metabolic rate and, consequently, decrease oxygen demand in SCIs where blood perfusion is already compromised. In rodent models of SCI, hypothermia has demonstrated the potential to enhance tissue preservation and facilitate functional recovery [175]. One-year outcomes from a phase I clinical trial investigating the effects of hypothermia in individuals with an AIS grade A SCI revealed no significant increase in complication rates when compared to the normothermic group [176].

The Miami Institutional Review Board (Central IRB) group is now conducting a prospective multicenter controlled trial on the use of modest systemic hypothermia for acute cervical SCIs (NCT02991690). In this study, they aim to study patients with acute cervical SCIs who received modest systemic hypothermia (33 °C) for 48 h in comparison with the standard medical care treatment. The preliminary data showed that hypothermia was not associated with an increased risk of complications within the first 6 weeks.

### 10.2. Neuroregenerative Therapies

#### 10.2.1. Pharmacological Therapies

##### NOGO-A Inhibition/Antibody

Nogo-A is a protein that is present in the CNS myelin and responsible for the inhibition of neurite growth [177,178]. In preclinical models, intrathecal administration of neutralizing monoclonal Anti-Nogo-A antibodies has shown promising results, promoting axonal regeneration recovery [178,179]. Freund et al. observed functional improvement, paralleled with histopathological findings, that suggested axonal sprouting.

A phase II trial (ClinicalTrials.gov: NCT0393353210) investigated changes in upper-extremity motor scores (UEMSs) according to the International Standards for the Neurological Classification of Spinal Cord Injury (ISNCSCI), after the administration of the antibody therapy was completed earlier this year, and the results are now being analyzed.

##### RGMa Inhibition with Human Monoclonal Antibodies

Repulsive guidance molecule A (RGMa) is found to be upregulated after an SCI and is known to play a role in neuronal apoptosis, as well as axonal growth inhibition and remyelination [180,181,182]. To neutralize this effect, monoclonal antibodies are being administered by investigators to limit the inhibitory action of RGMa.

Studies have shown improvements in the recovery of motor function and gait after its administration, reassuring neuronal survival and plasticity on descending serotonergic pathways and corticospinal tract axons [180,181,182]. In a rodent model of SCI, systemic or intrathecal anti-RGMa administration enhanced corticospinal tract regeneration and promoted neuronal survival and plasticity [1,2].

Two phase 2 clinical trials are now investigating Elezanumab (ClinicalTrials.gov: NCT04295538) and MT-3921 (ClinicalTrials.gov: NCT04683848) as the inhibitor for the RGMa, and they are likely to have their results published soon.

#### 10.2.2. Cell-Based Therapies

One of the most exciting areas of research in neuroregenerative therapies for SCIs involves cell-based interventions [183]. Stem cells, such as embryonic stem cells, induced pluripotent stem cells, and adult stem cells, have shown remarkable potential for regenerating damaged neural tissue. These cells can differentiate into various neural cell types and provide structural support, release growth factors, and modulate the inflammatory response [184].

##### Neural Stem Cells (NSCs)

Neural stem cells are self-renewing multipotent cells that have the ability to differentiate into various cell types found in the nervous system, including oligodendrocytes and astrocytes [185,186]. Transplantation of NSCs into the injured spinal cord has demonstrated the potential to promote axonal regeneration, remyelination, and the formation of functional neural connections in large animal models [184,187,188]. NSCs can also modulate the inflammatory response and provide trophic support to the damaged tissue.

In a paired phase I/II clinical trial performed by StemCells Inc. (Newark, CA, USA), human CNS stem cells were transplanted in thoracic AIS grade A–C SCI patients (NCT01321333) and cervical AIS grade B or C (NCT02163876) SCI patients, with the results revealing no safety concerns [189]. A one-year follow-up interim analysis of the cervical SCI patient cohort revealed a noticeable inclination towards motor improvements within the treatment group [190]. Due to financial limitations, the trial was prematurely terminated, resulting in underpowered results. However, it is worth mentioning that six years of follow-up data on the thoracic SCI cohort showed no significant safety concerns [191].

It is important to mention oligodendrocyte precursor cells (OPCs), which can be derived from NSCs. These cells possess a propensity for differentiating into myelinating oligodendrocytes and have garnered significant attention in translational research endeavors. In 2018, the SCiStar trial, a phase I/IIa dose-escalation study, focused on intramedullary AST-OPC1 injections in patients with SCI AIS grade A or B injuries ranging from C4 to C7. The trial’s outcomes, published last year, demonstrated the safe administration of LCTOPC1 to participants during the subacute phase following a cervical SCI. The injection procedure, along with the low-dose temporary immunosuppression regimen and utilization of LCTOPC1, exhibited favorable tolerability. The safety data, along with the neurological-function outcomes, provide substantial support for further investigation aimed at assessing the efficacy of LCTOPC1 in SCI treatment [192].

##### Mesenchymal Stem Cells (MSCs)

Mesenchymal stem cells (MSCs) are multipotent cells that can be obtained from various sources, such as bone marrow, adipose tissue, or umbilical cord tissue [193,194]. MSCs have immunomodulatory properties and can secrete a range of factors that promote tissue repair and reduce inflammation through immunomodulation, neurotrophic-factor secretion, and pro-angiogenic signaling [195,196,197]. Transplanted MSCs have shown the ability to enhance tissue preservation, stimulate endogenous regeneration mechanisms, and improve functional outcomes in SCI models. In a recent metanalysis of clinical trials, no major safety concerns were demonstrated. However, patients showed no significant benefit in terms of motor improvements with this therapy [198].

There are three ongoing phase II clinical trials investigating the safety and efficacy of umbilical-cord MSC transplantation in patients with subacute, early, and late stages of chronic SCI classified as AIS grades A-D (NCT03521336, *n* = 84; NCT03521323 *n* = 66; NCT03505034 *n* = 43). However, the outcomes of these trials have not yet been published.

##### Schwann Cells

Schwann cells are peripheral-nervous-system glial cells that play a crucial role in nerve regeneration. Transplantation of Schwann cells into the injured spinal cord has shown positive effects on axonal regrowth, myelination, and functional recovery. These cells can create a growth-promoting environment for regenerating axons and provide structural support to the injured tissue [199,200,201,202].

It is worth noting that cell-based therapies face challenges such as cell survival, integration into the host tissue, and guiding their differentiation into the desired cell types. Further research and clinical trials are required to optimize the cell types, delivery methods, and timing of cell transplantation to maximize their therapeutic potential in treating SCIs [185,203,204,205].

##### Olfactory Ensheathing Cells

Olfactory ensheathing cells (OECs) are a specialized type of glial cell found in the olfactory system and are known for their high phagocytic abilities [199,202,206]. They also possess unique regenerative properties, including the ability to promote axonal growth and remyelination [155,156,157]. OEC transplantation into the injured spinal cord has shown promise in promoting neural regeneration, improving locomotor function, and facilitating neuronal connectivity across the lesion site.

A systematic review and meta-analysis encompassing 62 experimental treatments involving 1164 animals demonstrated significant neurobehavioral recovery following the transplantation of OECs in an SCI [207]. Encouraged by these promising preclinical findings, several clinical trials have been conducted, with a meta-analysis incorporating 1193 patients. Although no significant increase in serious adverse events (SAEs) was observed after OEC transplantation in chronic SCIs, the efficacy of the treatment could not be definitively determined due to methodological considerations in the analyzed studies [208]. Consequently, further trials involving larger cohorts and randomized controlled trials are warranted to establish the efficacy of OEC transplantation in the context of SCIs.

### 10.3. Biomaterial Scaffolds

Biomaterial scaffolds provide a supportive environment for axonal regrowth and neural-tissue regeneration. These scaffolds can be made from natural or synthetic materials and are designed to mimic the extracellular matrix of the spinal cord. They provide physical support, guidance cues, and controlled release of therapeutic agents. Biomaterial scaffolds have shown promise in facilitating axonal regeneration, bridging the lesion site, and promoting tissue remodeling. However, challenges remain in achieving optimal integration with the host tissue and promoting functional connectivity. Amongst the therapies involved in this group, we can cite the following:

#### 10.3.1. QL6

This is a water-soluble biomaterial that self-assembles into a nanometer-scale lattice-like conformation. It can be delivered prior to or in addition to NSCs [203,204], and it works on enhancing graft survival, reducing inflammation and glial scarring, and improving motor function in animal models [206,209,210].

#### 10.3.2. Hyaluronan/Methylcellulose (HAMC)

HAMC is a biodegradable polymer blend [211] that works on the support of NSCs in SCIs and can be modified to carry and deliver growth factors [203,204] (e.g., PDGF-AA—platelet-derived growth factor AA), scar-degrading drugs [203,204] (e.g., chondroitinase ABC), and peptide ligands (e.g., RGD).

#### 10.3.3. Functional Electrical Stimulation

Functional electrical stimulation (FES) holds great potential as a non-invasive or implantable intervention for individuals with SCIs. FES is a therapeutic approach that uses electric currents to activate nerves and muscles to restore or improve functional movements. It can be applied to various target areas, such as muscles, peripheral nerves, or the spinal cord itself [144,148,212,213].

The electrical stimuli applied to these structures will work on depolarizing excitable tissues, generating muscle contractions or activating sensory pathways. FES can be used to improve upper- and lower-limb functions, as well as to increase bladder and bowel control, and can potentially be explored to improve respiratory function when applied directly to the respiratory muscles, even assisting with coughing, deep breathing, and clearing secretions [144,148,212,213].

**Table 1 children-10-01456-t001:** Summary of experimental clinical research and future treatments.

Neuroprotective strategies
1.1 Pharmacological therapies Riluzole is a benzothiazole sodium channel blocker that acts as a glutamatergic modulator [93,163,165,167]. Granulocyte colony-stimulating factor (*G-CSF*) is a cytokine glycoprotein that suppresses neuronal and oligodendroglial apoptosis [173,174]. 1.2 Nonpharmacological therapies Systemic hypothermia reduces the basal metabolic rate, decreasing the oxygen demand in SCIs where blood perfusion is compromised [176].
2. Neuroregenerative therapies
2.1 Pharmacological therapies NOGO-A inhibition/antibody aims to suppress the NOGO-A, which is a protein that is present in the CNS myelin that inhibits neurite growth [178,179]. *RGMa inhibition* aims to suppress RGMa, which is upregulated after an SCI and that plays a role in neuronal apoptosis [180,181,182]. 2.2 Cell-based therapies Neural stem cells [189,190,191], oligodendrocyte precursor cells (OPCs) [192], mesenchymal stem cells [198], Schwann cells [185,203,204,205], and olfactory ensheathing cells [207].
3. Biomaterial scaffolds QL6 is a water-soluble biomaterial that works on enhancing graft survival, reducing inflammation and glial scarring [210]. Hyaluronan/methylcellulose (HAMC) is a biodegradable polymer blend that works on the support of NSCs in SCIs and can be modified to carry and deliver growth factors [203,204].
4. Functional electrical stimulation Uses electric currents to activate nerves and muscles to restore or improve functional movements [144,148,212,213].

RGMa = repulsive guidance molecule A; G-CSF = granulocyte colony-stimulating factor.

## 11. Conclusions

In conclusion, while significant strides have been made in the field of SCI research, it is still notable that the pediatric population is yet to benefit from substantial clinical trials targeting this challenging condition. Research and therapeutic interventions have predominantly focused on adult populations, and the unique anatomical, physiological, and developmental characteristics of children’s spines necessitate targeted investigations to address their distinct needs.

It is crucial to emphasize that we recognize the lack of studies and data concerning the pediatric population with SCIs. Throughout this manuscript, we extensively acknowledged this limitation in each section. Our focus primarily centers on knowledge acquired within the adult population and its potential applicability to children. We hope that, in the near future, more studies will be directed to this population and that we can share in the opportunity to keep expanding the studies in this field.

## Figures and Tables

**Figure 1 children-10-01456-f001:**
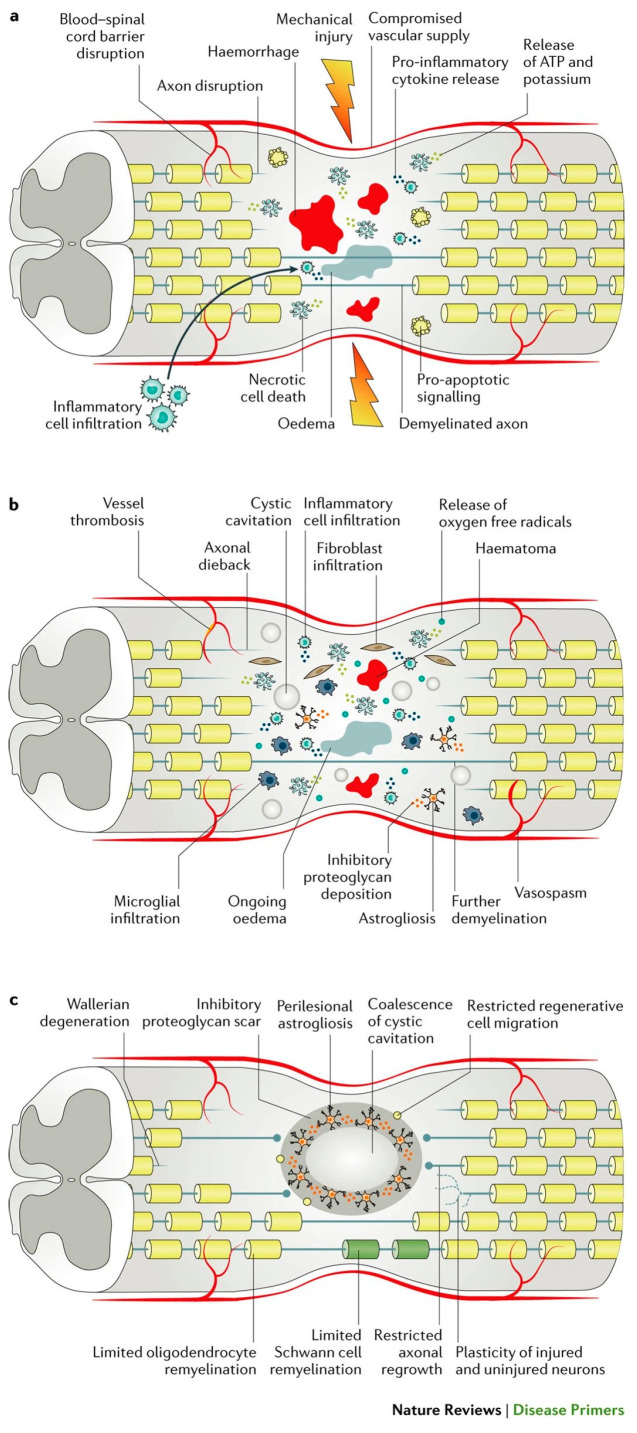
The pathophysiology of traumatic spinal cord injury. (**a**) Acute phase of a traumatic SCI (0–48 h post-injury). (**b**) Subacute phase (2–4 days post-injury). (**c**) Intermediate-to-chronic phase (2-weeks-to-6-months post-injury). Reprinted with permission from Ref. [32] Ahuja et al. (2017).

## Data Availability

We state that the manuscript was written in compliance with the Publication Ethics Guidelines. All primary sources of data are cited in the text.

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
