# Peer review of "Pediatric Spinal Cord Injury: A Review"

_children, 2023, doi:10.3390/children10091456_

Round 1

Reviewer 1 Report

Please see attached word document for comments/suggestions.

Please see attached word document for comments/suggestions.

Author Response

Reviewer #1:

Please see below for minor edits and clarifications

Response: We thank the reviewer for their review and comments.

2.1 Contrast to adults

-5 sacral vertebrae number is missing

Response: Thank you for noticing this. We have added the number as suggested.

3.3 Pathophysiology

-Figure 1, fix the pixel. The image is blurry.

Response: We have uploaded a better-quality version of the image.

4.1 4.1. X-ray, CT(A), MRI

-performed by instead of “perfume” by Viccelio

Response: We have corrected this error.

-Please reword this statement (confusing) Line 299

 “Usage of CT studies is not definite to clear every cervical spine in the great majority of this populations”

Response: We have reworded this sentence to enhance clarity.

-CT, AOD, MRI, and AANS/CNS- expand abbreviations when mentioned for the first time.

Response: We have expanded all abbreviations at first mention.

-Please reword this statement  Line 310 “tend to have worst poor outcome”. Should be worst outcomes or poorest outcomes.

Response: We have reworded this sentence to enhance clarity. 

Reviewer 2 Report

Thank you for submitting your review on Pediatric Spinal Cord Injury. You are to be congratulated on reducing essentially a textbook of information on this topic into a single review. There are only two issues that potentially need attention:

(1). One of the major reasons for the increased incidence of upper cervical spine fractures and associated spinal cord issues in younger children is the increased size of their head compared to their torso and the effect it can have the forces associated with motor vehicle accidents, falls and abuse. This is an important issue to pediatric orthopaedic spine surgeons. With maturation the normal ratio between head size and the torso is achieved with the shift of these forces to the lower cervical spine. This also the reason that transport boards in ambulances and pediatric emergency departments usually have boards with a hole for the occiput fit to decrease neck flexion in children with suspected cervical spine and spinal cord injuries. 

(2). In pediatric orthopaedics there is also a significant trend away from using CT scans during evaluation and treatment of cervical and other spinal injuries because of the later risk for radiation induced malignancies. This is substantial and related to the number of scans and total radiation exposure. It is possible to obtain CT scans with significantly reduced radiation. This needs to be stated in your review.

Author Response

Thank you for submitting your review on Pediatric Spinal Cord Injury. You are to be congratulated on reducing essentially a textbook of information on this topic into a single review. There are only two issues that potentially need attention:

Response: We thank the reviewer for their review and comments.

(1). One of the major reasons for the increased incidence of upper cervical spine fractures and associated spinal cord issues in younger children is the increased size of their head compared to their torso and the effect it can have the forces associated with motor vehicle accidents, falls and abuse. This is an important issue to pediatric orthopaedic spine surgeons. With maturation the normal ratio between head size and the torso is achieved with the shift of these forces to the lower cervical spine. This also the reason that transport boards in ambulances and pediatric emergency departments usually have boards with a hole for the occiput fit to decrease neck flexion in children with suspected cervical spine and spinal cord injuries. 

Response: We have considered the crucial concern regarding the increased head size of the pediatric population and have provided a more comprehensive explanation of the pathophysiology of spinal cord injury within this specific group.

(2). In pediatric orthopaedics there is also a significant trend away from using CT scans during evaluation and treatment of cervical and other spinal injuries because of the later risk for radiation induced malignancies. This is substantial and related to the number of scans and total radiation exposure. It is possible to obtain CT scans with significantly reduced radiation. This needs to be stated in your review.

Response: Thank you for highlighting this point. We have addressed this in the text.

It is also important to state that CT scan radiation is an important topic to address regarding the pediatric population, since this group has a higher risk for malignancies than adults [1,2]. Due to that risk, the new trend is to have reduced radiation CT scans or to evaluate the need for a CT scan at all [ 1,2]. This is a very nuanced call to be made, as there is a risk of missing diagnosis because of suboptimal image quality as a consequence of exposure settings being too low [1], so the discussion should always be towards minimizing risks for the patients.

Reviewer 3 Report

Dear authors, thank you for allowing me to review this interesting manuscript review on pediatric spinal cord injuries. I found the manuscript well-written and organized, with a clear structure from epidemiology to future treatment.

Although I found this manuscript of interest to the international readership, there are some points of concern I would like to raise to improve the already high quality of your work.

Major concerns:

- The most important flaw I have identified is that there is a complete lack of a methods paragraph in which an explanation of how you found the articles, what was your search strategy, what databases you have used, etc..., has been provided.

- There is a total lack of a limitations section.

Minor concerns:

- In the introduction, despite the sentences you have written before, you have provided the first citation on line 77.

- Page 10, Lines 459-462. You have cited the Spinal Cord Independence Measure as relevant to focus on restoring daily functions. However, a new instrument to assess self-care in spinal cord injuries (SC-SCII) has been recently developed, that should be tested on a pediatric population. Please read this "Conti A, Campagna S, Nolan M, Scivoletto G, Bandini B, Borraccino A, Vellone E, Dimonte V, Clari M. Self-care in spinal cord injuries inventory (SC-SCII) and self-care self-efficacy scale in spinal cord injuries (SCSES-SCI): development and psychometric properties. Spinal Cord. 2021 Dec;59(12):1240-1246. doi: 10.1038/s41393-021-00702-9. Epub 2021 Aug 28. PMID: 34455422."

- Page 14, Lines 633-634. I have clinical experience with SCI, but you might explain to a novice reader why spasticity could be desirable.

I found the manuscript well-written and organized. 

Just a few minor suggestions/typos:

Page 1, Line 29. "Intricacies" sounds weird, please replace it

Page 13, Line 574. Please, remove the symbol <=

Page 13, Line 609. Please, consider to replace "diagnose" with "diagnosis 

Author Response

Dear authors, thank you for allowing me to review this interesting manuscript review on pediatric spinal cord injuries. I found the manuscript well-written and organized, with a clear structure from epidemiology to future treatment.

Although I found this manuscript of interest to the international readership, there are some points of concern I would like to raise to improve the already high quality of your work.

Response: We thank the reviewer for their review and comments.

Major concerns:

- The most important flaw I have identified is that there is a complete lack of a methods paragraph in which an explanation of how you found the articles, what was your search strategy, what databases you have used, etc..., has been provided.

Response: We have added a section detailing the methods for this review.

- There is a total lack of a limitations section.

Response: We have added a paragraph within the concluding section discussing the limitations of our manuscript.

Minor concerns:

- In the introduction, despite the sentences you have written before, you have provided the first citation on line 77.

Response: Thank you for the observation, we have added citations to enhance our introduction.

- Page 10, Lines 459-462. You have cited the Spinal Cord Independence Measure as relevant to focus on restoring daily functions. However, a new instrument to assess self-care in spinal cord injuries (SC-SCII) has been recently developed, that should be tested on a pediatric population. Please read this "Conti A, Campagna S, Nolan M, Scivoletto G, Bandini B, Borraccino A, Vellone E, Dimonte V, Clari M. Self-care in spinal cord injuries inventory (SC-SCII) and self-care self-efficacy scale in spinal cord injuries (SCSES-SCI): development and psychometric properties. Spinal Cord. 2021 Dec;59(12):1240-1246. doi: 10.1038/s41393-021-00702-9. Epub 2021 Aug 28. PMID: 34455422."

Response: Recently, an instrument for spinal cord injuries inventory (SC-SCII) and self-care self-efficacy scale (SCSES-SCI) was developed for the adult population [1]. It has been shown to be a valid and reliable instrument for assessing self-care in adult patients, and this can be further tested and validated in the younger patients. This has been added to the text.

- Page 14, Lines 633-634. I have clinical experience with SCI, but you might explain to a novice reader why spasticity could be desirable.

Response: Managing spasticity must be individualized since some degree of it can be tolerable. This has been added to the manuscript.

Comments on the Quality of English Language

I found the manuscript well-written and organized. 

Just a few minor suggestions/typos:

Page 1, Line 29. "Intricacies" sounds weird, please replace it

Response: We have adjusted this wording.

Page 13, Line 574. Please, remove the symbol <=

Response: We have removed the symbol.

Page 13, Line 609. Please, consider to replace "diagnose" with "diagnosis 

Response: We have updated the word to diagnosis.

Reviewer 4 Report

we read with interest the article by CUNHA et al discussing Pediatric Spinal Cord Injury: involving the etiology and epidemiology of SCI in children focusing on the anatomical and physiological characteristics of the developing spinal cord along with the diagnostic methods available.  the authors conclude with the recent emerging research and innovative therapies in the field of  SCI. 

comments;

this work is going from a prominent group in the area of SCI led by Dr. Fehling, the article is well written and articulated especially in the first sections, however, I noticed that whole sections are written with several paragraphs without any references which need o be corrected and factually statement should be referenced, as this work is not an opinion work.

Second, certain sections should be presented in a table that is missing from this work these sections include the diagnostic part along with the mechanism involved as the figure presented is so small and hard to read what is inserted.

the major section that needs to be focused on is "emerging research and innovative therapies " which I think is the most important section of the work as previous parts have been already discussed in other reviews.

the authors should elaborate on these innovative therapies and present them with more details in a separate table if these are clinical studies coupled with outcomes and what labs conducted them such as intervention with stem cells, it should be detailing what ste cells and what clinical studies have been performed, what is with other drug intervention etc...

overall it is an excellent article to read

Author Response

we read with interest the article by CUNHA et al discussing Pediatric Spinal Cord Injury: involving the etiology and epidemiology of SCI in children focusing on the anatomical and physiological characteristics of the developing spinal cord along with the diagnostic methods available.  the authors conclude with the recent emerging research and innovative therapies in the field of  SCI.

Response: We thank the reviewer for their review and comments.

comments;

this work is going from a prominent group in the area of SCI led by Dr. Fehling, the article is well written and articulated especially in the first sections, however, I noticed that whole sections are written with several paragraphs without any references which need o be corrected and factually statement should be referenced, as this work is not an opinion work.

Response: Thank you for the observation, we have added citations to enhance the text.

Second, certain sections should be presented in a table that is missing from this work these sections include the diagnostic part along with the mechanism involved as the figure presented is so small and hard to read what is inserted.

Response: We also have uploaded a better version of this image.

the major section that needs to be focused on is "emerging research and innovative therapies " which I think is the most important section of the work as previous parts have been already discussed in other reviews.

Response: We have dedicated effort to enhancing the section on emerging research and innovative therapies, incorporating the most prominent and up-to-date studies in this field. We have also included a new section on rehabilitation, where we briefly explore innovative approaches and potential advancements in this area.

the authors should elaborate on these innovative therapies and present them with more details in a separate table if these are clinical studies coupled with outcomes and what labs conducted them such as intervention with stem cells, it should be detailing what ste cells and what clinical studies have been performed, what is with other drug intervention etc...

Response: We have uploaded a table summarizing the studies we have added as mentioned previously.

overall it is an excellent article to read

Response: Thank you

Round 2

Reviewer 3 Report

Dear authors, thank you for allowing me to review the second version of this interesting manuscript.

Although I found a general improvement in the overall quality of data presented, your study continues to lack a structured presentation of the review process, which must include the research design, search strategy, keywords, search strings, inclusion/exclusion criteria, data extraction, and synthesis.

Thus, in my position, I cannot be sure of the process of conducting the literature review, given the impossibility of replicating it. 

Author Response

Dear authors, thank you for allowing me to review the second version of this interesting manuscript.

Although I found a general improvement in the overall quality of data presented, your study continues to lack a structured presentation of the review process, which must include the research design, search strategy, keywords, search strings, inclusion/exclusion criteria, data extraction, and synthesis.

Thus, in my position, I cannot be sure of the process of conducting the literature review, given the impossibility of replicating it. 

>>Response: Thank you for the constructive feedback. We have further expanded our Methods section to detail the methods employed for this combined scoping and narrative review. This was a challenging review to undertake due to the broad scope of the topic (that the journal Children requested us to cover) as well as a paucity of literature on the topic. We formed a multidisciplinary team with expertise in translational spinal cord injury research, paediatric neurology, and paediatric neurosurgery to ensure a comprehensive review on the topic of pediatric spinal cord injury. The revised methods section is as follows:

This study employs a combined scoping and narrative review methodology. Our approach included a comprehensive electronic database search across PubMed, PubMed central (PMC), ScoPus, Google Scholar and the Cochrane Library. To ensure a thorough exploration of the topic, we employed a variety of search terms, namely: "Children OR Pediatric AND Spinal Cord Injury," "Pediatric AND Spinal Cord Injury AND Surgery OR clinical treatment," "Children AND Spinal Cord Injury AND early surgery," "Children OR Pediatric AND Spinal Cord Injury AND ICU management OR complications management," "Children OR Pediatric AND rehabilitation," and "Spinal cord injury AND new trials OR future treatment." Subsequently, we retrieved the identified articles and applied a set of rigorous inclusion criteria.

The inclusion criteria encompassed studies that were published in the English language, spanned various primary study designs and trials, and covered the spectrum of literature from the databases' inception to the review's cutoff date (February 15th, 2023). Our assessment extended to studies examining any form of pediatric spinal cord injury, irrespective of its origin (e.g., traumatic, non-traumatic, congenital). The scope of eligible studies encompassed diverse outcomes pertaining to pediatric spinal cord injuries, including but not limited to clinical presentations, treatment modalities, strategies for rehabilitation, long-term repercussions, quality of life assessments, and complications arising from the condition.

Reviewer 4 Report

Accept in present form

Author Response

No changes requested.